# Better transfer learning with inferred successor maps

**Tamas J. Madarasz**
University of Oxford
`tamas.madarasz@ndcn.ox.ac.uk`

**Timothy E. Behrens**
University of Oxford
`behrens@fmrib.ox.ac.uk`

## Abstract

Humans and animals show remarkable flexibility in adjusting their behaviour when their goals, or rewards in the environment change. While such flexibility is a hallmark of intelligent behaviour, these multi-task scenarios remain an important challenge for machine learning algorithms and neurobiological models alike. We investigated two approaches that could enable this flexibility: factorized representations, which abstract away general aspects of a task from those prone to change, and nonparametric, memory-based approaches, which can provide a principled way of using similarity to past experiences to guide current behaviour. In particular, we combine the successor representation (SR), that factors the value of actions into expected outcomes and corresponding rewards, with evaluating task similarity through clustering the space of rewards. The proposed algorithm inverts a generative model over tasks, and dynamically samples from a flexible number of distinct SR maps while accumulating evidence about the current task context through amortized inference. It improves SR's transfer capabilities and outperforms competing algorithms and baselines in settings with both known and unsignalled rewards changes. Further, as a neurobiological model of spatial coding in the hippocampus, it explains important signatures of this representation, such as the "flickering" behaviour of hippocampal maps, and trajectory-dependent place cells (so-called splitter cells) and their dynamics. We thus provide a novel algorithmic approach for multi-task learning, as well as a common normative framework that links together these different characteristics of the brain's spatial representation.

## 1 Introduction

Despite recent successes seen in reinforcement learning (RL) [1, 2], some important gulfs remain between sophisticated reward-driven learning algorithms, and the behavioural flexibility observed in biological agents. Humans and animals seem especially apt at the efficient transfer of knowledge between different tasks, and the adaptive reuse of successful past behaviours in new situations, an ability that has sparked renewed interest in machine learning in recent years.

Several frameworks have been proposed to help move the two forms of learning closer together, by incorporating transfer and generalisation capabilities into RL agents. Here we focus on two such ideas: abstracting away the general aspects of a family of tasks and combining it with specific task features on the fly through factorisation [3, 4]. And nonparametric, memory-based approaches [5, 6, 7, 8] that may help transfer learning by providing a principled framework for reusing information, based on inference about the similarity between the agent's current situation, and situations observed in the past.

We focus in particular on a specific instance of the transfer learning problem, where the agent acts in an environment with fixed dynamics, but changing reward function or goal locations (but see section 5 for more involved changes in a task). This setting is especially useful for developing an intuition about how an algorithm balances the retention of knowledge about the environment shared between tasks, while specializing its policy for the current instantiation at hand. This is also a central challenge

in the related problem of continual learning that has been examined in terms of stability-plasticity trade-off [9], or catastrophic forgetting [10, 11].

Dayan's SR [3] is well-suited for transfer learning in settings with fixed dynamics, as the decomposition of the value function into representations of expected outcomes (future state occupancies) and corresponding rewards allows us to quickly recompute values under new reward settings. Importantly, however, SR also suffers from limitations when applied in transfer learning scenarios: the representation of expected future states still implicitly encodes previous reward functions through its dependence on the behavioural policy under which it was learnt, which in turn was tuned to exploit these previous rewards. This can make it difficult to approximate the new optimal value function following a large change in the environment's reward function, as states that are not en route to previous goals/rewards will be poorly represented in the SR. In such cases the agent will stick to visiting old reward locations that are no longer the most desirable, or take suboptimal routes to new rewards [12, 13].

To overcome this limitation, we combine the successor representation with a nonparametric clustering of the space of tasks (in this case the space of possible reward functions), and compress the representation of policies for similar environments into common successor maps. We provide a simple approximation to the corresponding hierarchical inference problem and evaluate reward function similarity on a diffused, kernel-based reward representation, which allows us to link the policies of similar environments without imposing any limitations on the precision or entropy of the policy being executed on a specific task. This similarity-based policy recall, operating at the task level, allows us to outperform baselines and previous methods in simple navigation tasks. Our approach naturally handles unsignalled changes in the reward function with no explicit task or episode boundaries, while also imposing reasonable limits on storage and computational complexity. Further, the principles of our approach should readily extend to settings with different types of factorizations, as SR itself can be seen as an example of generalized value functions [4] that can extend the dynamic programming approach usually applied to rewards and values to other environmental features.

We also aim to build a learning system whose components are neuroscientifically grounded, and can reproduce some of the empirically observed phenomena typical of this type of learning. The presence of parallel predictive representations in the brain has previously been proposed in the context of simple associative learning in amygdala and hippocampal circuits [14, 15], as well as specifically in the framework of nonparametric clustering of experiences into latent contexts using a computational approach on which we also build [16]. Simultaneous representation of dynamic and diverse averages of experienced rewards has also been reported in the anterior cingulate cortex [17] and other cortical areas, and a representation of a probability distribution over latent contexts has been observed in the human orbitofrontal cortex [18]. The hippocampus itself has long been regarded as serving both as a pattern separator, as well as an autoassociative network, with attractor dynamics enabling pattern completion [19, 20]. This balance between generalizing over similar experiences and tasks by compressing them into a shared representation, while also maintaining task-specific specialization is a key feature of our proposed hippocampal maps.

On the neurobiological level we thus aim to offer a framework that binds these ideas into a common representation, linking two putative, but disparate functions of the hippocampal formation: a prospective map of space [21, 22, 23], and an efficient memory processing organ, in this case compressing experiences to help optimal decision making. We simulate two different rodent spatial navigation tasks: in the first we show that our model gives insights into the emergence of fast, "flickering" remapping of hippocampal maps [24, 25], seen when rodents navigate to changing reward locations [26, 27]. In the second task, we provide a quantitative account of trajectory-dependent hippocampal representations (so-called splitter cells) [21] during learning. Our model therefore links these phenomena as manifestations of a common underlying learning and control strategy.

## 2   Reinforcement Learning and the Successor Representation

In RL problems an agent interacts with an environment by taking actions, and receiving observations and rewards. Formally, an MDP can be defined as the tuple $\mathcal{T} = (\mathcal{S}, \mathcal{A}, p, \mathcal{R}, \gamma)$, specifying a set of states $\mathcal{S}$, actions $\mathcal{A}$, the state transition dynamics $p(s'|s, a)$, a reward function $\mathcal{R}(s, a, s')$, and the discount factor $\gamma \in [0, 1]$, that reduces the weight of rewards obtained further in the future. For the

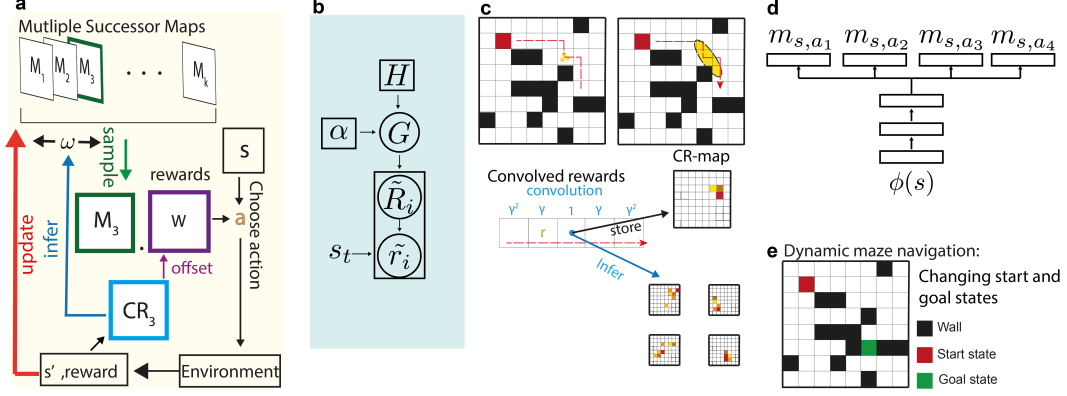

Figure 1: Model components:(a) Overview of the model. A successor map/network is sampled according to $\omega$, the probability weight vector over contexts. This sampled map is used to select an action one or more SR receives TD updates, while $\omega$ is also updated given the experienced reward, using inference in a generative model of expected rewards. (b) Dirichlet process Gaussian mixture model of these average, or convolved reward (CR) values. The Dirichlet process is defined by a base distribution $H$ and concentration parameter $\alpha$, giving a distribution over CR value distributions. (c) Computing CR maps by convolving discounted rewards along experienced trajectories. (d) Neural network architecture for continuous state space tasks.(e) Example navigation environment for our experiments.

transfer learning setting, we will consider a family of MDPs with shared dynamics but changing reward functions, $\{\mathcal{T}(\mathcal{S}, \mathcal{A}, p, \cdot)\}$, where the rewards are determined by some stochastic process $\mathscr{R}$.

Instead of solving an MDP by applying dynamic programming to value functions, as in Q-learning [28], it is possible to compute expected discounted sums over future state occupancies as proposed by Dayan's SR framework. Namely, the successor representation maintains an expectation over future state occupancies given a policy $\pi$:

$$M_t^\pi(s, a, s') = \mathbb{E}(\sum_{k=0}^{\infty} \gamma^k \mathbb{I}_{(s_{t+k+1}=s')}|s_t = s, a_t = a) \tag{1}$$

We will make the simplifying assumption that rewards $r(s, a, s')$ are functions only of the arrival state $s'$. This allows us to represent value functions purely in terms of future state occupancies, rather than future state-action pairs, which is more in line with what is currently known about prospective representations in the hippocampus [22, 23, 29]. Our proposed modifications to the representation, however, extend to the successor representation predicting state-action pairs as well.

In the tabular case, or if the rewards are linear in the state representation $\phi(s)$, the SR or successor features can be used to compute the action-value function $Q(s, a)$ exactly, given knowledge of the current reward weights $\mathbf{w}$ satisfying $r(s, a, s') = \phi(s') \cdot \mathbf{w}$. In this case the SR and the value function are given by

$$M_t^\pi(s, a, :) = \mathbb{E}(\sum_{k=0}^{\infty} \gamma^k \phi(s_{t+k+1})|s_t = s, a_t = a), \quad Q_t^\pi(s, a) = \sum_{s'} M_t^\pi(s, a, s') \cdot \mathbf{w}(s'). \tag{2}$$

We can therefore apply the Bellman updates to this representation, as follows, and reap the benefits of policy iteration.

$$M(s_t, a_t, :) \leftarrow M(s_t, a_t, :) + \alpha(\phi(s_{t+1}) + \gamma \cdot M(s_{t+1}, a^\star, :) - M(s_t, a_t, :))$$
$$a^\star = arg \max_a (M(s_{t+1}, a, :) \cdot \mathbf{w}) \tag{3}$$

# 3 Motivation and learning algorithm

## 3.1 Model setup

Our algorithm, the Bayesian Successor Representation (BSR, Fig. 1a, Algorithm 1 in Supplementary Information) extends the successor temporal difference (TD) learning framework in the first instance by using multiple successor maps. The agent then maintains a belief distribution $\omega$, over these maps, and samples one at every step, according to these belief weights. This sampled map is used to generate an action, one (e.g. the most likely, or the sampled SR) or all SR maps receive TD updates, while the reward and observation signals are used to perform inference over $\omega$.

Our approach rests on the intuition that it is advantageous to transfer policies between task settings with similar reward functions, where similarity in this case means encountering similar goals or rewards along the similar trajectories in state space. The aim is to transfer policies between environments where similar rewards/goals are near each other, while avoiding negative transfer, and without relying on model-based knowledge of the environment's dynamics that can be hard to learn in general and can introduce errors. To achieve this, BSR adjudicates between different successor maps using online inference over latent clusters of reward functions. We evaluate reward similarity using a kernel-based average of rewards collected along trajectories, a type of local (though temporally bidirectional) approximation to the value function. These average, or convolved reward (CR) values $\mathbf{v}^{cr}$, (Fig. 1c) are then used to perform inference using a nonparametric Dirichlet process mixture model [30]. The mixture components determining the likelihoods of the $\mathbf{V}^{cr}$ are parametrized using a CR map for each context, represented by the vector $\mathbf{w}_i^{cr}$, giving a Gaussian linear basis function model.

$$G \sim DP(\alpha, H) \qquad\qquad CR\_map_i = \mathbf{W}_i^{cr} \sim G \qquad (4)$$

$$V_i^{cr}(s) \sim \mathcal{N}(\phi(s) \cdot \mathbf{W}_i^{cr}, \sigma_{CR}^2) \qquad\qquad\qquad\qquad\qquad\qquad (5)$$

We regard the choice of successor map as part of the inference over these latent contexts, by attaching, for each context, a successor map $M_i$ to the corresponding CR map $\mathbf{w}_i^{cr}$, which allows us to use the appropriate $M_i$ for a policy iteration step (3), and for action selection

$$a = \arg\max_a \left( M_i(s_t, a, :) \cdot \mathbf{w}_t \right). \qquad (6)$$

Namely, we sample $M_i$ from the distribution over contexts, $\omega$ to choose an action while pursuing a suitable exploration strategy, and perform weighted updates either on all maps if updates are inexpensive (e.g in the tabular case) or only the most likely map $M_{argmax(\omega)}$ or the sampled map $M_i$ if updates are expensive (e.g. function approximation case). Finally, we define the observed CR values as a dot product between rewards received and a symmetric kernel of exponentiated discount factors $K_\gamma = [\gamma^{-f}, \ldots, \gamma^f]^T$, such that $v_i^{cr}(s) = \mathbf{r}_{t-f:t+f} \cdot K_\gamma$. The common reward feature vector $\mathbf{w}$ is learned by regressing it onto experienced rewards, i.e. minimizing $\|\phi(s_t)^T \cdot \mathbf{w} - r(t)\|$. This setup allows the agent to accumulate evidence during and across episodes, infer the current reward context as it takes steps in the environment, and use this inference to improve its policy and select actions.

## 3.2 Inference

Inference in this setting is challenging for a number of reasons: like value functions, CR values are policy dependent, and will change as the agent improves its policy, even during periods when the reward function is stationary. Since we would like the agent to find the optimal policy for the current environment as quickly as possible, the sampling will be biased and the $v^{cr}$ observation likelihoods will change. Further, the dataset of observed CR values expands with every step the agent takes, and action selection requires consulting the posterior at every step, making usual Markov Chain Monte Carlo (MCMC) approaches like Gibbs-sampling computationally problematic.

Sequential Monte Carlo (SMC) methods, such as particle filtering, offer a compromise of faster computation at the cost of the 'stickiness' of the sampler, where previously assigned clusters are not updated in light of new observations. We adopt such an approach for inference in our algorithm, as we believe it to be well-suited to this multi-task RL setup with dynamically changing reward functions and policies. The interchangeability assumption of the DP provides an intuitive choice for

the proposal distribution for the particle filter, so we extend each particle $\mathbf{c}^i$ of partitions by sampling from the prior implied by the Chinese Restaurant Process (CRP) view of the DP:

$$P(c_t^i = k \mid \mathbf{c}_{1:t-1}^i) = \begin{cases} \frac{m_k}{t-1+\alpha}, \text{ where } m_k \text{ is the number of observations assigned to cluster } k \\ \frac{\alpha}{t-1+\alpha} \text{ , if } k \text{ is a new cluster} \end{cases}$$

(7)

The importance weight for a particle of context assignments $p_i$ requires integrating over the prior (base) distribution $H$ using all the CR value observations. If we adopt a conjugate prior to the Gaussian observation likelihood in (5), this can be done analytically: for a multivariate Gaussian prior for the mixture components the posterior CR maps and the posterior predictive distribution required to calculate the importance weights will both be Gaussian.

This procedure, which we term Gaussian SR (GSR) still requires computing separate posteriors over the CR maps for each particle, with each computation involving the inversion of a matrix with dimensions $dim(\phi)$ (See S1 for details). We therefore developed a more computationally efficient alternative, with a single posterior for each CR map and a single update performed on the map corresponding to the currently most likely context. We also forgo the Gaussian representation, and performed simple delta rule updates, while annealing the learning rate. Though we incur the increased space complexity of using several maps, by limiting computation to performing TD updates, and approximate inference outlined above, BSR provides an efficient algorithm for handling multi-task scenarios with multiple SR representations.

Recently proposed approaches by Barreto et al. also adjudicate between several policies using successor features [31, 32] and as such we directly compare our methods to their generalized policy iteration (GPI) framework. In GPI the agent directly evaluates the value functions of all stored policies, to find the overall largest action-value, and in later work also builds a linear basis of the reward space by pre-learning a set of base tasks. This approach can lead to difficulties with the selection of the right successor map, as it depends strongly on how sharply policies are tuned and which states the agent visits near important rewards. A sharply tuned policy for a previous reward setting with reward locations close to current rewards could lose out to a broadly tuned, but mostly unhelpful competing policy. On the other hand, keeping all stored policies diffuse, or otherwise regularising them can be costly, as it can hinder exploitation or the fast instantiation of optimal policies given new rewards. Similarly, constructing a set of base tasks [32] can be difficult as it might require encountering a large number of tasks before successful transfer could be guaranteed, as demonstrated most simply in a tabular setting.

## 4 Experiments

### 4.1 Grid-world with signalled rewards and context-specific replay

We first tested the performance of the model in a tabular maze navigation task (Fig. 1e), where both the start and goal locations changed every 20 trials, giving a large number of partially overlapping tasks. This was necessary to test for precise, task-specific action selection in every state, which is not required under some other scenarios [31, 12]. In the first experiment, the reward function was provided to directly test the algorithms' ability to map out routes to new goals. Episodes terminated when the agent reached the goal state and received a reward, or after 75 steps if the agent has failed to reach the goal. We added walls to the interior of the maze to make the environment's dynamics non-trivial, such that a single SR representing the uniform policy (diffusion-SR) would not be able to select optimal actions. We compared BSR to a single SR representation (SSR), an adaptation of GPI (SFQL) from Barreto et al. [31] to state-state SR, as well as an agent that was provided with a pre-designed clustering, using a specific map whenever the goal was in a particular quadrant of the maze (Known Quadrant, KQ). Each algorithm, except SSR, was provided with four maps to use, such that GPI, once all its maps were in use, randomly selected one to overwrite, otherwise following its original specifications. We added a replay buffer to each algorithm, and replayed randomly sampled transitions for all of our experiments in the paper. Replay buffers had specific compartments for each successor map, with each transition added to the compartment corresponding to the map used to select the corresponding action. The replay process thus formed part of our overall nonparametric approach to continual learning. We tested each algorithm with different $\epsilon$-greedy exploration rates $\epsilon \in [0., 0.05, \ldots, 0.35]$ (after an initial period of high exploration) and SR learning rates $\alpha_{SR} \in [0.001, 0.005, 0.01, 0.05, 0.1]$. Notably BSR performed best across all

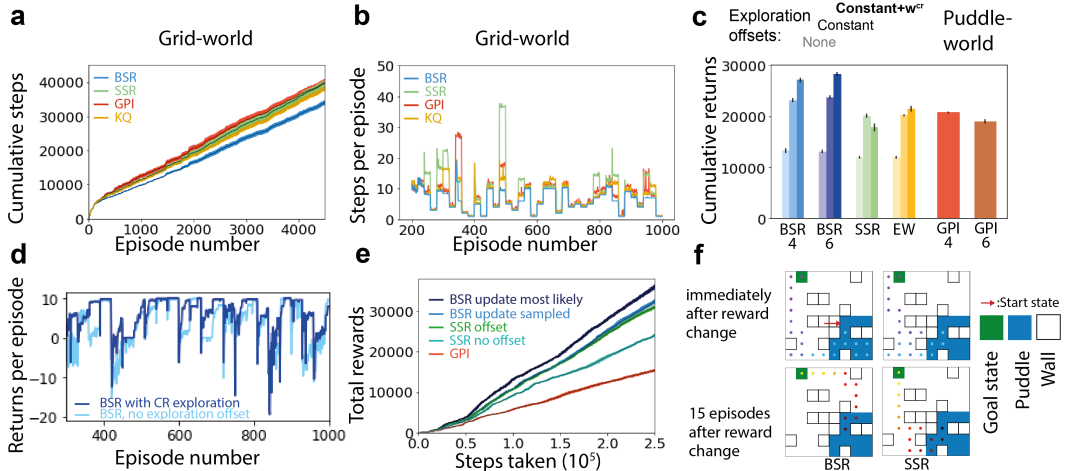

Figure 2: Simulation results for best parameter settings. Error bars represent mean ± s.e.m. (a) Total number of steps taken across episodes to reach a changing, but signalled goal in the tabular grid-world navigation task.(b) BSR adjusts better to new goals than the other algorithms, as illustrated by the average length of each episode. (c) BSR with two types of exploration bonuses performs best at navigation with unsignalled goals, puddles, and task-boundaries in a puddle-world environment. Different shades represent different exploration offsets for the relevant algorithms. (d) Exploration bonuses help BSR find new goals faster. (e) The proposed improvements transfer to a navigation task in a continuous state-space maze with unsignalled goal locations and task boundaries. (f) Example trajectories from Experiment II showing how BSR, but not SSR adjusts to the optimal route over the same episodes.

parameter settings, but performed best with $\epsilon = 0$, whereas the other algorithms performed better with higher exploration rates. The total number of steps taken to complete all episodes, using the best performing setting for each algorithm, is shown in Fig. 2a and Table S1. Figs. 2b and S2 show lengths of individual episodes. Increasing the number of maps in GPI to 10 led to worse performance by the agent, showing that it wasn't the lack of capacity, but the inability to generalize well that impaired its performance. We also compared BSR directly with the matched full particle filtering solution, GSR, which performed slightly worse (Fig. S1), suggesting that BSR could maintain competitive performance with decreased computational costs.

## 4.2 Puddle-world with unsignalled reward changes and task boundaries

In the second experiment, we made the environment more challenging by introducing puddles, which carried a penalty of 1 every time the agent entered a puddle state. Reward function changes were also not signalled, except that GPI still received task change signals to reset its use of the map and reset its memory buffer. Negative rewards are known to be problematic and potentially prevent agents from properly exploring and reaching goals, we therefore tried exploration rates up to and including 0.65, and also evaluated an algorithm that randomly switched between the different successor maps at every step, corresponding to BSR with fixed and uniform belief weights (Equal Weights, EW). This provided a control showing that it was not increased entropy or dithering that drove the better performance of BSR (Fig. 2c).

## 4.3 Directed exploration with reward space priors and curiosity offsets

As expected, optimal performance required high exploration rates from the algorithms in this task (Table S.2), which afforded us the opportunity to test if CR maps could also act to facilitate exploration. Since they act as a kind of prior for rewards in a particular context, it should be possible to use them to infer likely reward features, and direct actions towards these rewards. Because of the linear nature of SR, we can achieve this simply by offsetting the now context-specific reward weight vector $\mathbf{w}_i$ using the CR maps $\mathbf{w}^{cr}$ (Algorithm 1, line 7). This can help the agent flexibly redirect its

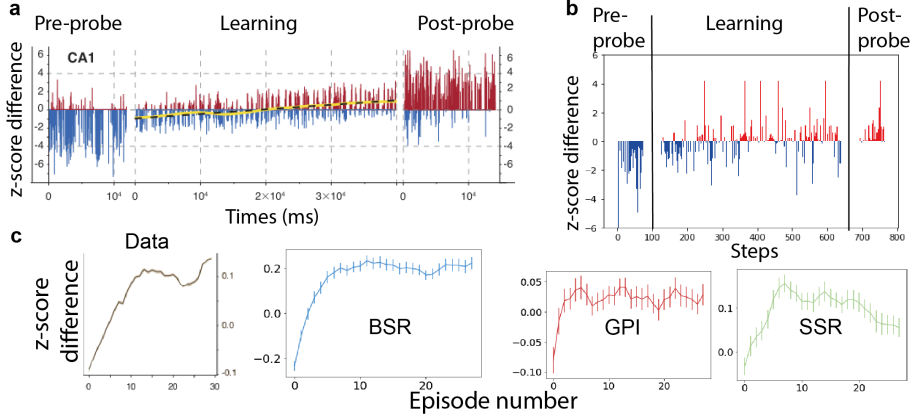

Figure 3: Multiple maps and flickering representations in the hippocampus. Fast paced hippocampal flickering as (a) animals, and (b) BSR adjust to a new reward environment. For every time step, vertical lines show the difference between the z-scored correlation coefficients of the current firing pattern to the pre-probe and to the post-probe firing patterns. (c) Average evolution of z-score differences across learning trials within a session.

exploration following a change in the reward function, as switching to a new map will encourage it to move towards states where rewards generally occur in that particular context (Fig. 2d,f). We also experimented with constant offsets to **w** before each episode, which in turn is related to upper confidence bound (UCB) exploration strategies popular in the bandit task setting. Under the assumption e.g. that reward features drift with Gaussian white noise between episodes, using a simple constant padding function gives an UCB for the reward features. While we saw strong improvements from these constant offsets for SSR and EW as well, BSR also showed further strong improvements when the two types of exploration offsets were combined. These offsets represent a type of curiosity in the multi-task setting, where they encourage the exploration of states or features not visited recently, but they are unlike the traditional pseudo rewards often used in reward shaping. They only temporarily influence the agent's *beliefs* about rewards, but never the actual rewards themselves. This means that this reward-guidance isn't expected to interfere with the agent's behaviour in the same manner as pseudo rewards can [33], however, like any prior, it can potentially have detrimental effects as well as the beneficial ones demonstrated here. We leave a fuller exploration of these, including applying such offsets stepwise and integrating it into approaches like GPI, for future work.

### 4.4 Function approximation setting

The tabular setting enables us to test many components of the algorithm and compare the emerging representations to their biological counterparts. However, it is important to validate that these can be scaled up and used with function approximators, allowing the use of continuous state and action spaces and more complex tasks. As a proof of principle, we created a continuous version of the maze from Experiment I, where steps were perturbed by 2D Gaussian noise, such that agents could end up visiting any point inside the continuous state space of the maze. State embeddings were provided in the form of Gaussian radial basis functions and agents used an artificial neural network equivalent of Algorithm 1, where the Matrix components $M(:, a, :)$ were replaced by a multi-layer perceptron $\psi_a$ (Fig. 1d). We tested BSR-4 with two different update strategies vs. SSR and GPI-4 in this setting, with exploration rates up to 0.45, with BSR outperforming the other two again. Fig. 2c shows the performance of the algorithms in terms of total reward collected by timestep in the environment, to emphasize the connection with continual learning.

## 5 Neural data analysis

### 5.1 Hippocampal flickering during navigation to multiple, changing rewards

Our algorithm draws inspiration from biology, where animals face similar continual learning tasks while foraging or evading predators in familiar environments. We performed two sets of analyses

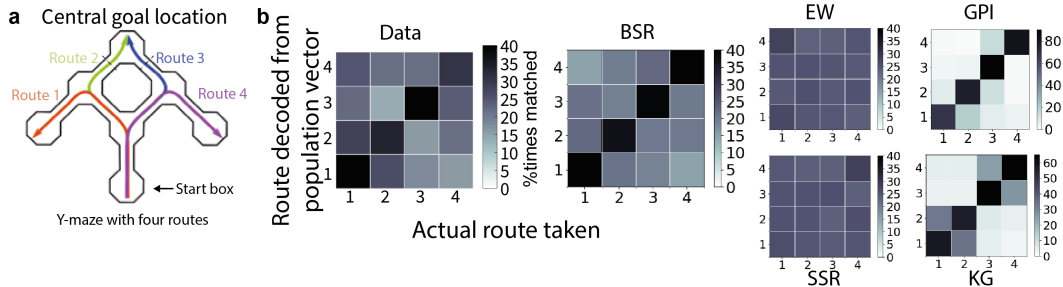

Figure 4: Splitter cell representations from animals and artificial agents completing a Y-maze navigation task with changing goals. (a) Outline of the Y-maze, adapted from [21]. Three possible goal locations define four possible routes. (b) Information about trial type in the animals' neural representation is similar to that in BSR. Horizontal axes show the actual trial type, vertical axes the decoded trial type at the start of the trial. Known Goal (KG) uses the same principle as KQ before.

to test if the brain uses similar mechanisms to BSR, comparing our model to experimental data from rodent navigation tasks with changing reward settings. We used the successor representation as a proxy to neural firing rates as suggested in [23], with firing rates proportional to the expected discounted visitation of a state $\mathbf{r}_t(s') \propto M(s_t, a_t, s')$. Our framework predicts the continual, fast-paced remapping of hippocampal prospective representations, in particular in situations when changing rewards increases the entropy of the distribution over maps. Intriguingly, such 'flickering' of hippocampal place cells have indeed been reported, though a normative framework accounting for this phenomenon has been missing. Dupret et al. and Boccara et al. [27, 26] recorded hippocampal neurons in a task where rodents had to collect three hidden food items in an open maze, with the locations of the three rewards changing between sessions. Both papers report the flickering of distinct hippocampal representations, gradually moving from being similar the old to being similar to the new one as measured by z-scored similarity (Fig. 3a, adapted with permission from [26]). BSR naturally and consistently captured the observed flickering behaviour as shown on a representative session in Fig. 3b (with further sessions shown in Figs. S6-S9). Further, it was the only tested algorithm that captured the smooth, monotonic evolution of the z-scores across trials (Fig. 3c), and gave the closest match of 0.90 for the empirically measured correlation coefficient of 0.95 characterizing this progression [26].

## 5.2 Splitter cells

Another intriguing phenomenon of spatial tuning in the hippocampus is the presence of so-called splitter cells that exhibit route-dependent firing and current location conditional both on previous and future states [34]. While the successor representation is purely prospective, in BSR the inference over the reward context depends also on the route taken, predicting exactly this type of past-and-future dependent representation. Further, rather than a hard-assignment to a particular map, our model predicts switching back and forth between representations in the face of uncertainty. We analysed data of rats performing a navigation task in a double Y-maze with 3 alternating reward locations [21] (Fig. 4a, adapted with permission). The central goal location has two possible routes (Route 2 and 3), one of which is blocked every time this goal is rewarded, giving 4 trial types. These blockades momentarily change the dynamics of the environment, a challenge for SR [13]. Our inference framework, however, overcomes this challenge by 'recognizing' a lack of convolved rewards along the blocked route when the animal can't get access to the goal location, allowing the algorithm to switch to using the alternate route. Other algorithms, notably GPI, struggles with this problem (Fig. S4). as it has to explicitly map the policy of going around the maze to escape the barrier. To further test our model's correspondence with hippocampal spatial coding, we followed the approach adopted in Grieves et al. [21] for trying to decode the trial type from the hippocampal activity as animals begin the trial in the start box. The analysis was only performed on successful trials, and thus a simple prospective representation would result in large values on the diagonal, as in the case of GPI and KQ. In contrast, the empirical data resembles the pattern predicted by BSR, where a sampling of

maps results in a more balanced representation, while still providing route dependent encoding that differentiates all four possible routes already in the start box.

## 6    Related work

A number of recent or concurrent papers have proposed algorithms for introducing transfer into RL/deep RL settings, by using multiple policies in some way, though none of them use an inferential framework similar to ours, provides a principled way to deal with unsignalled task boundaries, or explains biological phenomena. We extensively discuss the work of Barreto et al. [31, 32] in the paper. Our approach shares elements with earlier work on probabilistic policy reuse [35], which also samples from a distribution over policies, however does so only at the beginning of each episode, doesn't follow a principled approach for inferring the weight distribution, and is limited int performance by its use of Q-values rather than factored representations. Wilson et al. [36] and Lazaric et al. [37] employed hierarchical Bayesian inference for generalization in multi-task RL using Gibbs sampling, however neither used the flexibility afforded by the successor representation or integrated online inference and control as we do in our method. [36] uses value iteration to solve the currently most likely MDP, while [37] applies the inference directly on state-value functions. Other approaches tackle the tension between generality and specialization by regularizing specialized policies in some way with a central general policy [38, 39], which we instead expressly tried to avoid here. General value functions in the Horde architecture [4] also calculate several value functions in parallel, corresponding to a set of different pseudo-reward functions, and in this sense are closely related, but a more generalized version of SR. Schaul et al. combined the Horde architecture with a factorization of value functions into state and goal embeddings to create universal value function approximators (UVFAs) [40]. Unlike our approach, UVFA-Horde first learns goal-specific value functions, before trying to learn flexible goal and state embeddings through matrix factorization, such that the successful transfer to unseen goals and states depends on the success of this implicit mapping. More recent work from Ma et al. [41] and Borsa et al. [42] combines the idea of universal value functions and SR to try to learn an approximate universal SR. Similarly to UVFAs however, [41] relies on a neural network architecture implicitly learning the (smooth) structure of the value function and SR for different goals, in a setting where this smoothness is supported by the structure in the input state representation (visual input of nearby goals). Further, this representation is then used only as a critic to train a goal-conditioned policy for a new signalled goal location. [42] proposes to overcome some of these limitations by combining the UVFA approach with GPI. However, it doesn't formulate a general framework for choosing base policies and their embeddings when learning a particular task space, or for sampling these policies, or addresses the issue of task boundaries and online adjustment of policy use. Other recent work for continual learning also mixed learning from current experience with selectively replaying old experiences that are relevant to the current task [43, 44]. Our approach naturally incorporates, though is not dependent, on such replay, where relevant memories are sampled from SR specific replay buffers, thus forming part of the overall clustering process. [45] also develops a nonparametric Bayesian approach to avoid relearning old tasks while identifying task boundaries for sequence prediction tasks, with possible applications for model-based RL, while [46] explored the relative advantages of clustering transition and reward functions jointly or independently for generalization. [12, 13] also discuss the limitations of SR in transfer scenarios, and [47] found evidence of these difficulties in certain policy revaluation settings in humans.

## 7    Conclusion and future work

In this paper we proposed an extension to the SR framework by coupling it with the nonparametric clustering of the task space and amortized inference using diffuse, convolved reward values. We have shown that this can improve the representation's transfer capabilities by overcoming a major limitation, the policy dependence of the representation, and turning it instead into a strength through policy reuse. Our algorithm is naturally well-suited for continual learning where rewards in the environment persistently change. While in the current setting we only inferred a weight distribution over the different maps and separate pairs of SR bases and CR maps it opens the possibility for approaches that create essentially new successor maps from limited experience. One such avenue is the use of a hierarchical approach similar to hierarchical DP mixtures [48] together with the composition of submaps or maplets which could allow the agent to combine different skills according to the task's demand. We leave this for future work. Further, in BSR we only represent uncertainty

as a distribution over the different SR maps, but it is straight forward to extend the framework to represent uncertainty within the SR maps (over the SR associations) as well, and ultimately to incorporate these ideas into a more general framework of RL as inference [49].

## 8 Acknowledgements

We would like to thank Rex Liu for his very detailed and helpful comments and suggestions during the development of the manuscript, as well as Evan Russek and James Whittington for helpful comments and discussions. Thanks also to Roddy Grieves and Paul Dudchenko for generously sharing data from their experiments.

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
