[Supplementary Material]

# Supplementary Information for 'Better transfer learning with inferred successor maps'

**Tamas J. Madarasz**
University of Oxford
tamas.madarasz@ndcn.ox.ac.uk

**Timothy E. Behrens**
University of Oxford
behrens@fmrib.ox.ac.uk

## S1 Algorithm details

### S1.1 Particle filtering with Bayesian Linear Gaussian Model

In this section we outline the particle filtering solution to assigning each CR value observation to a cluster in the conjugate prior case. The DP mixture model can be written as the infinite limit of the following hierarchical model as $K \to \infty$[1]

$$\mathbf{p} \sim \text{Dirichlet}(\alpha/K, \dots, \alpha/K) \tag{1}$$

$$\varphi_c \sim H \tag{2}$$

$$c_i | \mathbf{p} \sim \text{Discrete}(\mathbf{p}) \tag{3}$$

$$y_i | c_i, \varphi \sim F(\varphi_{c_i}) \tag{4}$$

where $y_i$ are the observations, and $F(\theta_i) = F((\varphi_{c_i})$ are the mixture components. Using the notation of our setup more directly, at the t-th step

$$\mathbf{p} \sim \text{Dirichlet}(\alpha/K, \dots, \alpha/K) \tag{5}$$

$$\mathbf{w}_c^{cr} \sim H \tag{6}$$

$$c_t | \mathbf{p} \sim \text{Discrete}(\mathbf{p}) \tag{7}$$

$$V_{(s_t)}^{cr} | c_t, \mathbf{w}^{cr} \sim \mathcal{N}(\phi(s_i) \cdot \mathbf{w}_{c_t}^{cr}, \sigma_{CR}^2) \tag{8}$$

When performing Gibbs sampling, the state of the Markov chain consists of the $c_t$-s and $\mathbf{w}_c^{cr}$s that are currently 'in use'. Similarly, the state of the particle filter is represented, for every particle, by an assignment of $c_t$-s, and the posteriors in use, $\mathbf{w}_c^{cr}$. If the base distribution $H$ represents a conjugate prior, such as a multivariate Gaussian

$$\mathbf{w}^{cr} \sim \mathcal{N}(M_0, \Sigma_0), \tag{9}$$

we can perform the integration analytically, giving a Gaussian posterior for $\mathbf{w}_{c_t}^{cr}$, and a Gaussian posterior predictive distribution to calculate the importance weight $w^t$. In more detail, given a particle of partitions $(c_1, \dots c_t)$ and observations $v_{1:t}^{cr}$,

$$\mathbf{w}_{c_t}^{cr} | H, v_{1:t}^{cr} \sim \mathcal{N}(M_t^{c_t}, \Sigma_t^{c_t}) \tag{10}$$

$$\Sigma_t^{c_t} = \left[ (\Sigma_{t-1}^{c_t})^{-1} + \frac{\phi(s_t)^T \phi(s_t)}{\sigma_{CR}^2} \right]^{-1} \tag{11}$$

$$M_t^{c_t} = \Sigma_t^{c_t} \left[ (\Sigma_{t-1}^{c_t})^{-1} \cdot M_{t-1}^{c_t} + \frac{\phi(s_t)^T v_t^{cr}}{\sigma_{CR}^2} \right] \tag{12}$$

The particle filter itself proceeds by first sampling from the proposal distribution given by the CRP

$$P(c_t^i = k \mid \mathbf{c}_{1:t-1}^i) = \begin{cases} \frac{m_k}{t-1+\alpha}, \text{ where } m_k \text{ is the number of observations assigned to cluster } k \\ \frac{\alpha}{t-1+\alpha}, \text{ if k is a new cluster} \end{cases}$$

$$\tag{13}$$

Figure S1: Comparable performance of GSR and BSR on Experiment I.

and then computing the importance weight given by the posterior predictive based on the observations up to and including t-1.

$$w_t^p \propto w_{t-1}^p * p(v_t^{cr}|c_t, \mathbf{c}_{1:t-1}, \mathbf{v}_{1:t-1}^{cr}), \text{where} \tag{14}$$

$$v_t^{cr}|c_t, \mathbf{c}_{1:t-1}, \mathbf{v}_{1:t-1}^{cr} \sim \mathcal{N}\left(\phi(s_t)^T M_{t-1}^{c_t}, \phi(s_t)^T \Sigma_{t-1}^{c_t} \phi(s_t) + \sigma_{CR}^2\right) \tag{15}$$

Since we resample at every step, in practice the importance weights were given simply by the above the densities, normalized across all particles,

$$\hat{w}_t^p = \frac{w_t^p}{\sum_i w_t^i}.$$

This procedure, used in our 'GSR' algorithm, requires at least one inversion of the precision matrix at every time step to compute the posterior, which is computationally intensive in the non-tabular case, or when the prior is not diagonal. For BSR, we forgo storing separate Gaussian posteriors of the CR maps for each particle, updating instead one common set of CR maps. At every step, after computing the CR value $v^{cr}$ and the importance weights $w_t^p$, but before resampling, the particular context (map) to be updated is chosen through a winner-take-all majority of the summed up aggregated normalized importance weights

$$arg \max_k \{ \sum_{p:c_t^p=k} \hat{w}_t^p \}.$$

Because we chose the CR kernel $K_\gamma = \gamma^{-f}, \gamma^{-f+1}, \ldots, 1, \ldots, \gamma^f$ to be symmetric in time, the particle filtering process only started once the agent has taken $f$ steps. Similarly, at the end of the episode, the last $f$ steps were 'filtered together', using a simple proposal, and importance weights calculated together using the last $f$ likelihoods. CR values for the last $f$ states were calculated by padding the vector of received rewards (and the normalizing vector for the tabular representation) by 0s at the end.

GSR and BSR performed similarly for Experiment I (Fig. S1), but GSR seemed to perform considerably worse on Experiment II, though we did not conduct a full search of the hyperparameters.

We used a second version of the algorithm BSR2 for the neural data analysis, where we performed the CR map updates only at the end of each episode, by cycling through the observed CR values once, and updating the map for the *overall* most likely context as determined by the $\omega$ computed at the end of the episode. The maps remained unchanged during the episode when they were used to compute the importance weights. This further reduced the amount of computation required during acting in the environment, and resulted in CR maps with more concentrated clusters of similar values, but had only a small effect on performance.

## S1.2   Tabular algorithm

Here we provide details for our agent not included in the main text. The concentration parameter for the Dirichlet process was $\alpha = 2$, the standard deviation for the Gaussian generative distribution was

$\sigma_{cr} = 1.6$ but these parameters were not properly optimized or otherwise systematically evaluated. $P$ was a 100 by 10 matrix, corresponding to 100 particles each with 10 of the most recent contexts. In the tabular setting, the learning rate $\alpha_w$ was 1, $\alpha_{ws} = 0.01$, and $\alpha_{cr}$ was annealed starting from 0.15, over 6000 episodes. The replay updates were done on 5 randomly sampled transitions from the replay buffer after every direct update, with each successor map assigned its own replay buffer. The replay didn't use information about whether states were terminal or not, using the full Bellman backup each time. For GPI the buffer for a new map was reset whenever a change in the environment was signalled, so as not to contaminate the learning of the current policy with transitions from a different task setting, as this proved considerably detrimental in the experiments we tried. The delay in filtering, $f$, was set to 3 steps. Exploration rate was annealed during the first 250 episodes from 1 to 0, but the annealing stopped whenever it reached the exploration rate specified in the algorithm's parameter setting. We searched through $\alpha_{SR}$ values in the range specified above, and values of $\epsilon$ in $[0., 0.05, 0.1, \ldots, 0.35]$ for Experiment I, $[0., 0.05, 0.1, \ldots, 0.65]$ for the algorithms in Experiment II with no offsets, and $[0., 0.05, 0.1, \ldots, 0.45]$ for algorithms with offsets and Experiment III with the continuous maze. Best-performing settings are summarized in the tables below. The constant offset $c_{ws}$ was 1 in both tabular and continuous experiments. The replay buffer for each context could store 300 transitions before the oldest entry was overwritten. Minibatches for the SR replay updates contained 5 transitions (or the total number of transitions in the buffer if that was smaller).

After arriving in a state $s$, CR values were calculated by taking the dot product of the kernel $K_\gamma$ and the rewards received up to 3 steps before, and up to 3 steps after the state, appropriately padded with 0s if necessary, and normalized by the dot product of $K_\gamma$ and a vector of 1s for rewards and 0s for any padding used.

## S1.3 GSR

We matched GSR to BSR as much as possible, with 100 particles each storing the last ten contexts. We experimented with other setups (fewer particles, longer window of past contexts), but didn't get any improvements. The prior covariance matrix $\Sigma_0$ was the identity matrix, and the prior mean $M_0$ was a vector of zeros.

## S1.4 Neural network algorithm

For the neural network architecture, the input layer had 100 neurons, hidden layer sizes were 150, while the output layers by definition had the same size as the input, but with a separate output layer for each of the four possible actions (Fig. 1d), resulting in a total of 400 neurons. We used the tanh function as nonlinearity, and no nonlinearity was used for the output layers. Parameters were initialised using Glorot initialisation [2]. We employed the successor equivalent of target networks [3] for better performance (see Algorithm 2), and a designated replay buffer accompanying each successor network. This was equivalent to a single memory with transitions labelled by sampled context, as the limit on the memory size, determining when a memory was overwritten, was by episode (set to 200 episodes), and thus common to all memories. Transitions were sampled randomly from all the transitions stored in the buffer, with minibatches of size 15 (or the number of transitions stored in the buffer if this was smaller), used to update the successor features.. In addition, we did a second run of updates for $\mathbf{w}$ at the end of each episode, cycling through all the steps of the episode.

Parameters were $\alpha_w = 0.005$, $\alpha_{sr} = 0.0005$, and $\alpha_{cr}$ was annealed from 0.005 to 0.001 over 4000 episodes. Replay updates were performed on minibatches of 15 transitions, one minibatch after every transition, following a direct update based on the most recent transition. The rmsprop [4] optimizer was used for updates, and a dropout rate of 0.1 was applied. Networks and target networks were synchronized every 80 steps, starting from the beginning of each episode. The delay in filtering, $f$, was set to 4 steps, and $P$ was a 100 by 50, matrix, storing the 50 most recent contexts.

We ran BSR and SSR using the settings under which they performed best in Experiment II, namely BSR with both constant and $\mathbf{w}^{cr}$ exploration offsets, and SSR+ with constant offset.

Figure S2: Episode lengths for the full length of Experiment I

| Algorithm | Steps($10^3$) | $\epsilon, \alpha_{SR}$ |
|---|---|---|
| BSR-4 | $34.1 \pm 0.7$ | 0., 0.005 |
| GPI-4 | $40.0 \pm 1.$ | 0.05, 0.001 |
| SSR | $39.8 \pm 0.9$ | 0.1, 0.001 |
| KQ | $38.5 \pm 0.9$ | 0.05, 0.001 |

Table S1: Results and parameter settings for Experiment I.

## S2 Environment and experiment details

### S2.1 Grid-world maze

We implemented the tabular algorithms using the rllab framework [5] The maze for Experiment I was, as depicted in Fig. 1e, a 8 x 8 tabular maze. Available actions were up, down, left and right. If, on taking an action, the agent hit an internal or external wall, it stayed in place. Reward on landing on the goal was 10, $\gamma = 0.99$, and episodes were restricted to at most 75 steps. Start and goal locations changed every 20 episodes. Each algorithm was run 10 times, and results appropriately averaged.

### S2.2 Puddle-world

For Experiment II, puddles were added in the quadrant opposite the current reward, covering the entire quadrant except where walls were already in place. Landing in a puddle carried a penalty of -1, and puddles stayed in place during the entire episode (i.e. couldn't be 'picked up'). Otherwise things remained unchanged from the previous setting, except that the reward function now changed every 30 episodes, for a total of 150 sessions. Each algorithm was run 10 times as above, and results appropriately averaged.

| | No offset | | | Constant offset | | | Constant+CR offset | |
|---|---|---|---|---|---|---|---|---|
| Algorithm | Returns($10^3$) | $\epsilon, \alpha_{SR}$ | | Returns($10^3$) | $\epsilon, \alpha_{SR}$ | | Returns($10^3$) | $\epsilon, \alpha_{SR}$ |
| BSR-4 | $13.3 \pm 0.5$ | 0.55, 0.01 | | $23.2 \pm 0.4$ | 0.15, 0.05 | | $27.1 \pm 0.5$ | 0.05, 0.05 |
| BSR-6 | $13.1 \pm 0.4$ | 0.55, 0.01 | | $23.8 \pm 0.3$ | 0.35, 0.05 | | $28.3 \pm 0.4$ | 0.1, 0.05 |
| SSR | $12.0 \pm 0.2$ | 0.6, 0.005 | | $20.1 \pm 0.4$ | 0.25, 0.005 | | $17.9 \pm 0.8$ | 0.15, 0.01 |
| EW-4 | $11.9 \pm 0.3$ | 0.55, 0.005 | | $20.2 \pm 0.2$ | 035, 0.05 | | $21.6 \pm 0.5$ | 0.15, 0.05 |

| | Stored reward map, no offset | |
|---|---|---|
| Algorithm | Returns($10^3$) | $\epsilon, \alpha_{SR}$ |
| GPI-4 | $20.1 \pm 0.2$ | 0.55, 0.01 |
| GPI-6 | $19.0 \pm 0.4$ | 0.5, 0.01 |

Table S2: Results and parameter settigns for Experiment II.

Figure S3: Continuous maze, with the agent navigating from the start state (blue) to the unsignalled goal location (green).

| Algorithm | Total Rewards($10^3$) | $\epsilon$ |
|---|---|---|
| BSR-4 update most likely | $36.1 \pm 0.5$ | 0.3 |
| BSR-4 update sampled | $32.4 \pm 0.4$ | 0.3 |
| GPI-4 | $15.4 \pm 0.3$ | 0.35 |
| SSR+ | $31.0 \pm 0.4$ | 0.4 |
| SSR | $24.1 \pm 0.3$ | 0.45 |

Table S3: Results and parameter settings for Experiment III.

## S2.3   Continuous maze

This was a continuous copy of the maze from Experiment I. We set the length of the maze to be 3 but the maze was partitioned into an equivalent 8 x 8 setting by identically arrange walls as before (Fig. S3). 100 input neurons represented the agent's location, each with respect to one of 100 equally spaced locations as a Gaussian likelihood with diagonal covariance of 0.1. The activation of the i-th unit, with center $c_x, c_y$ was

$$\phi_i(x, y) \propto \exp\left(-\frac{(c_x - x)^2 + (c_y - y)^2}{2\sigma^2}\right),$$

where the normalizing constant was the same as for the Gaussian distribution in question, multiplied by 10. The actual state representation was the sum of the current activation, and the discounted sum of past states with a discount factor of 0.9, adding an element of trajectory-dependent recurrence and variability to the state representation.

Actions were again up, down, left, or right, but the arrival point of the step was offset by two-dimensional Gaussian noise, with a diagonal covariance matrix with 0.02 on the diagonals. The agent's step-size was 0.3 (thus smaller than before, at a tenth of the maze's length). Agents were point-like, but were not allowed to touch, or traverse walls. Steps that would have resulted in such an outcome instead meant that the agent stayed in place. This was also the case if the addition of the random noise would have resulted in contact between the agent and a wall. The agent collected the reward and ended the episode if it was at a distance of less than 0.25 from the goal's location. The goal location itself could be anywhere outside the walled-off areas of the maze. As in Experiment II, the reward function changed every 30 episodes. In all other aspects the task was identical to Experiment I.

Figure S4: Difference in performance between GPI and BSR in a task where the environmental dynamics temporarily change.

### S2.4    Three-reward open maze foraging task

Here the environment was the 8 by 8 maze with no internal walls. 3 reward locations were sampled at the beginning of each of the 150 sessions that lasted 30 trials (episodes) each. The pre- and post-probe parts of the sessions were 75 steps each (the same number as the upper limit of steps for an episode), and for simplicity we assumed no learning during these times. Instead the agent was randomly moving around while preserving its SR and CR representations. We used the values of the successor maps to represent firing rates, with $r_t(s') = M(s_t, a_t, s')$. For data analysis, the pre- and post-probe averages were calculated by averaging over the successor maps according to their weights at the beginning of the probe $\sum_i \omega_i M_i$. Since rewards in this task are not Markovian, and implementing a working memory or learning the overall structure of the task was not our focus, we gave all agents the ability to block out and temporarily set to 0 in $\mathbf{w}$ all rewards they already collected on that trial when evaluating the value function. This did not otherwise affect their beliefs about where the rewards were, i.e. they didn't forget learnt reward locations by the next trial. The algorithms' parameters were $\epsilon = 0.2$, $\alpha_w = 0.5$, $\alpha_{sr} = 0.1$ and $\sigma_{cr} = 1$. We got the following values for the Spearman correlation coefficients between the trial number and the z-score difference, for trials 1 to 16, averaged across sessions:

| BSR | SSR | EWI | GPI |
|---|---|---|---|
| $0.900 \pm 0.020$ | $0.778 \pm 0.030$ | $-0.033 \pm 0.045$ | $0.753 \pm 0.047$ |

The first 25 sessions were used as a warm-up for the agent to learn the environment, and sessions 25 to 140 were used in the data analysis. Plots equivalent to that in Fig. 3b for every 10th session from session 30 onwards are shown in figures S3 to S6 for the different algorithms.

### S2.5    Y-maze navigation task

We also implemented the Y-maze on a grid-world, as shown in Figure S2. On trial types 2 and 3 one of the two barriers shown were put in place, and the goal was in the top left hand corner state. The top right and top left hand corners were equivalent states (state 1, equivalent to the top goal state of the Y-maze in Fig. 4a), such that if the agent entered the top right hand corner, it got teleported to the top left and that was the only destination state recorded. Similarly, if it took the 'right' action from the top left corner it ended up in the state below the top right corner. This was permissible, since we don't assume any type of action embedding, and there is no generalization based on action identity. This setup thus gave us an equivalent representation to the Y-maze used in rodent experiments.

We ran the simulations on the exact session structure followed during the experiments and the agent had to complete the same number of successful episodes in each session that the experimental animals have done. In addition, we gave the agents a 500 episode pre-training phase, before we started following the experimental trial structure and 'recording' the representations. During these first 500 episodes, the trial type was reset every 20 episodes, as in Experiment I. After the pre-training phase there were 24 blocks of sessions where each block contained consecutive trials of all the trial types,

such that there were exactly 3 changes to the trial type during every block. We used the same blocks of trial types as used for the experiments in [6]. The algorithms' parameters were $\epsilon = 0.2$, $\alpha_w = 0.5$, $\alpha_{sr} = 0.1$ and $\sigma_{cr} = 1$.

## S3  Statistical Analysis

We performed a one-way ANOVA and a Tukey post-hoc test for Experiment I for the total number of steps taken to complete the 4500 episodes. The ANOVA showed an overall significant difference $F = 9.87$, $p < 10^{-5}$, and post-hoc Tukey's HSD test confirmed that BSR was significantly different from the other 3 algorithms. For Experiment III, we ran the same tests on the total rewards collected after 250000 steps. The overall ANOVA was again significant, $F = 354.3$, $p < 10^{-5}$, and BSR-4 with full exploration offsets was also significantly better than SSR and GPI according to Tukey's HSD test. Tukey's HSD test didn't find a significant difference when making the post-hoc comparison between BSR-4 updating the sampled context and SSR+, but BSR-4 updating the most likely context was significantly better than all other algorithms. Sample sizes were 10 for all algorithms, except BSR, for which it was 14.

## S4 Algorithms

---

**Algorithm 1** Bayesian Successor Representation

---

1: Require: discount factor $\gamma$, max number of clusters $k$, filter-delay $f$, hyperparameters $\{\epsilon, \sigma_{cr}, \alpha, c_{ws}, \alpha_{sr}, \alpha_{cr}, \alpha_w\}$
2: Initialise Successor Maps $M_1, .., M_k$ to 0, CR maps $\mathbf{w}_1^{cr}, .... \mathbf{w}_k^{cr}$, and weights $\mathbf{w}_1, .... \mathbf{w}_k$ to small random values, $\omega \leftarrow \{1/k, ..., 1/k\}$, random particle matrix $\mathbf{P} \sim$ Multinomial$(\omega)$, empty memory buffers $MB_1, \ldots, MB_k$. $K_\gamma \leftarrow [\gamma^{-f}, \gamma^{-f+1}, .., 1, .., \gamma^f]^T$
3: **for** each episode **do**
4:     $t \leftarrow 0$, initial state $s \leftarrow s_0$
5:     **while** $s$ not terminal and steps taken in episode<limit **do**
6:         $i \sim$ Multinomial$(\omega)$                             $\triangleright$ sample context with probabilites $\omega$
7:         $\mathbf{w}_j \leftarrow \mathbf{w}_j + \alpha_{ws}(c_{ws} + \mathbf{w}_j^{cr})\forall j$
8:         random_action $\sim$ Bernoulli$(\epsilon)$                      $\triangleright$ $\epsilon$-greedy exploration
9:         **if** not random_action **then**
10:             $a \leftarrow \underset{a}{arg\ max}\ M_i(s, a, :) \cdot \mathbf{w}_i$
11:         **else**
12:             a $\sim$ Uniform$(\{1,...,|A|\})$
13:         **end if**
14:         Execute $a$ and obtain next state $s'$ and reward $r = r(s')$
15:         Store $(s, a, s', r)$ in $MB_i$
16:         $\mathbf{w}_j \leftarrow \mathbf{w}_j + \alpha_w[r - \phi(s')^T \mathbf{w}_j]\phi(s')$ for all $j$
17:         **for** each context i **do**
18:             $a' \leftarrow \underset{a}{arg\ max}\ M_i(s', a, :)^T \mathbf{w}_i$
19:             $M_i(s, a, :) \leftarrow M_i(s, a, :) + \frac{\alpha_{sr}}{MC} * [\phi(s') + \gamma M_i(s', a', :) - M_i(s, a, :)]$
20:             Optional replay updates on mini-batch from $MB_i$
21:         **end for**
22:         **if** steps taken in episode $\geq$ f **then**
23:             $v_{t-f}^{cr} \leftarrow ([r_{\max(t-2*f,1)} : r_t] \cdot K_\gamma)/\sum K_\gamma$         $\triangleright$ compute normalized CR-value
24:             Filter$(\mathbf{P}, s_{t-f}, v_{t-f}^{cr})$
25:             $i \leftarrow argmax(\omega)$
26:             $\mathbf{w}_i^{cr} \leftarrow \mathbf{w}_i^{cr} + \alpha_{cr}[cr_t - \phi(s_t)^T \mathbf{w}_i^{cr}]\phi(s_t)$
27:         **end if**
28:         $s \leftarrow s'$
29:     **end while**
30: **end for**
31: **function** FILTER$(\mathbf{P}, s, v^{cr})$
32:     **for** every particle p (row of $\mathbf{P}$) **do**
33:         generate new context $c^p$ according to CRP prior (Eq. 20) with concentration parameter $\alpha$
34:     **end for**
35:     **for** every context i present in the proposals **do**
36:         Evaluate importance weights using Gaussian likelihoods $w^i = f_G(v^cr|\phi(s)^T \mathbf{w_i^{cr}}, \sigma_{cr}^2)$
37:     **end for**
38:     Resample particles according to $\hat{w}^p = \frac{w^{c^p}}{\sum_p w^{c^p}}$
39:     Remove first (oldest) column of $\mathbf{P}$
40:     Recompute $\omega$: $\omega[i] \leftarrow \sum_{p:c^p=i} \hat{w}^p\ \forall i$
41: **end function**

---

---
**Algorithm 2** Neural Bayesian Successor Representation
---
1: Require: discount factor $\gamma$, max number of clusters $k$, filter-delay $f$, state embedding $\phi$, hyperparameters $\{\epsilon, \sigma_{cr}, \alpha, c_{ws}, \alpha_{sr}, \alpha_{cr}, \alpha_w\}$

2: Initialise successor network and target network parameters $\theta_1, \theta_1^- .., \theta_k, \theta_k^-$ for networks $m_1, m_1^-, ....m_k, m_k^-$, weights $\mathbf{w}_1, ....\mathbf{w}_k$ and CR maps $\mathbf{w}_1^{cr}, ....\mathbf{w}_k^{cr}$ with small random values, $\omega \leftarrow \{1/k, ..., 1/k\}$, random particle matrix $\mathbf{P} \sim \text{Multinomial}(\omega)$, empty memory buffers $MB_1, \ldots, MB_k$.

3: $K_\gamma \leftarrow [\gamma^{-f}, \gamma^{-f+1}, .., 1, .., \gamma^f]$

4: **for** each episode **do**

5:      $t \leftarrow 0$, initial state $s \leftarrow s_0$

6:      **while** $s$ not terminal and steps taken in episode$<$limit **do**

7:          $i \sim \text{Multinomial}(\omega)$            $\triangleright$ sample context with probabilities $\omega$

8:          Every n_sync steps, synchronize $\theta_i$ and $\theta_i^- \forall i$

9:          $\mathbf{w}_j \leftarrow \mathbf{w}_j + \alpha_{ws}(c_{ws} + \mathbf{w}_j^{cr}) \forall j$          $\triangleright$ reward weight offset by CR prior

10:         random_action $\sim \text{Bernoulli}(\epsilon)$          $\triangleright$ $\epsilon$-greedy exploration

11:         **if** not random_action **then**

12:             $a \leftarrow arg\,\max\limits_{a} m_i(s,a) \cdot \mathbf{w}_i$

13:         **else**

14:             $a \sim \text{Uniform}(\{1,...,|A|\})$

15:         **end if**

16:         Execute $a$ and obtain next state $s'$ and reward $r = r(s')$

17:         Store $(s, a, s', r)$ in $MemoryBuffer_i$

18:         $\mathbf{w}_j \leftarrow \mathbf{w}_j + \alpha_w[r - \phi(s')^T \mathbf{w}_j]\phi(s') \; \forall \; j$

19:         $i \leftarrow arg\,\max\limits_{i} (\omega_i)$

20:         Sample a mini-batch of transitions from $MB_i$

21:         **for** each transition $s_m, a_m, s_m'$ **do**

22:             $a_m' \leftarrow arg\,\max\limits_{a} m_i^-(s_m', a_m) \cdot \mathbf{w}_i$

23:             perform updates $\theta_i \leftarrow \theta_i + \alpha_{sr} \nabla_{\theta_i}[\phi(s_m') + \gamma \cdot m_i^-(s_m', a_m') - m_i(s_m, a_m)]$

24:         **end for**

25:         **if** steps taken in episode $\geq f$ **then**

26:             $v_{t-f}^{cr} \leftarrow [r_{\max(t-2*f,1)} : r_t] \cdot K_\gamma$          $\triangleright$ compute CR-value

27:             Filter$(\mathbf{P}, s_{t-f}, v_{t-f}^{cr})$

28:             $\mathbf{w}_i^{cr} \leftarrow \mathbf{w}_i^{cr} + \alpha_{cr}[v_t^{cr} - \phi(s_t)^T \mathbf{w}_i^{cr}]\phi(s_t)$

29:         **end if**

30:         $s \leftarrow s'$

31:      **end while**

32: **end for**

33: **function** FILTER$(\mathbf{P}, s, v^{cr})$

34:      **for** every particle p (row of $\mathbf{P}$) **do**

35:          generate new context $c^p$ according to CRP prior (Eq. 20) with concentration parameter $\alpha$

36:      **end for**

37:      **for** every context i present in the proposals **do**

38:          Evaluate importance weights using Gaussian likelihoods $w^i = f_G(v^c r | \phi(s)^T \mathbf{w_i^{cr}}, \sigma_{cr}^2)$

39:      **end for**

40:      Resample particles according to $\hat{w}^p = \frac{w^{c^p}}{\sum_p w^{c^p}}$

41:      Remove first (oldest) column of $\mathbf{P}$

42:      Recompute $\omega$: $\omega[i] \leftarrow \sum_{p:c^p=i} \hat{w}^p \; \forall i$

43: **end function**
---

Figure S5: Splitter cell Y-maze

Figure S6: BSR flickering sessions

Figure S7: EW flickering sessions

Figure S8: SSR flickering sessions

Figure S9: GPI flickering sessions