[Reviews · NeurIPS 2019]

Reviewer 1



It is a very interesting and timely study which applies an intuitive (though non-trivial) idea of using multiple successor representation maps in reinforcement learning and adjudicating between them based on evidence coming from the environment. This is very relevant for understanding human and animal behaviour in complex environments with changing task conditions and reward contingencies. As this (and digital phenotyping more generally) gain increasing popularity, modelling and understanding these processes has increasing importance. Although the idea is fairly straightforward, I believe it has not been done before, hence the study is original. Literature is reviewed properly, appropriate and interesting analyses are performed, hence quality and significance are high as well. The greatest weakness of this paper in its current form is clarity, which hopefully can be improved, as although successor representation is an increasingly popular area in RL, it's also fairly complicated, hence it needs to be well explained (as e.g. is done in Gershman, J. Neurosci 2018). I also have a few more technical comments: - It's not exactly clear where is reward in section 2. Tabular case and rewards being linear in the state representation are mentioned; however, how exactly this is done should be explained more explicitly (or at least referred to where it's explained in the supplementary information - currently SI has information about parameters, algorithm and task settings details, but not methodological explanations) - It is mentioned that the mixture model is learned by gradient descent - it would be nice to see further discussion about how exactly this is done and why that is biologically realistic (as gradient descent is not something typically performed in the brain). - It would be nice to see not only summary statistics, but also typical trajectories performed by the model (and other candidate models) at different stages of learning - It is mentioned that epsilon = 0 works best for BSR, but in section 4.2 it's stated that for the puddle world epsilon = 0.2 was used for all models - why is that? Normally when comparing different models/algorithms, effort should be taken to find the best performing parameters (or more generally most suitable formalisations) for each model. - What exactly is the correlation coefficient in section 5.1 (0.90 or 0.95) between? - In Fig. 4, is it possible that GPI with noise added could reproduce the data similarly well or are there other measures to show that GPI cannot have as good fit with behavioural data (e.g. behavioural trajectories? time to goal?) - Finally this approach seems to be suitable for modelling pattern separation tasks, for which there is also behavioural data available - it would be nice to have some discussion on this. - There are a number of typos throughout the paper, which although don't obscure meaning should be corrected in the final version.

Reviewer 2



The paper presents a novel idea, is overall clearly written, and presents an interesting contribution. However the paper falls short in the following aspects which need to be addressed before publishing. Figure 2 shows experiments where the reward function changes every 20 trails. According to my reading, the “Single SR” baseline experiment is almost identical to the transfer experiments presented by Lehnert et al 2017 (https://arxiv.org/pdf/1708.00102.pdf). Why does the performance of at least the “Single SR” baseline not degrade right after a signaled reward change? Rather than reporting cumulative steps, the results would be much clearer by reporting per episode steps or per episode returns and comparing the actual convergence rates of all tested algorithm. When making these performance comparisons, which learning rates where tested? Was a grid search pass performed? Performing a gridsearch pass over a range of learning rates, exploration settings, etc. is important for ensuring reliability of the presented results. This should be done at least for the tabular experiments. Further, it would help to benchmark/compare simulations that do not use the CR-map rewards and analyze how exploration degrades/changes. I think the idea of convolving the reward map is interesting, but according to my reading the paper does not empirically support why this is necessary. Other related papers studying the dependency of the SR on a policy at transfer are https://www.nature.com/articles/s41562-017-0180-8 and https://journals.plos.org/ploscompbiol/article?id=10.1371/journal.pcbi.1005768. These two papers should also be discussed, as the submission attempts to improve over these previous findings.

Reviewer 3



I'm very concerned with the clarity of the paper. Many notations are used without definition and the method is not clearly described. For example, w(s) in L98 and phi(s') in L101 are used without definition, and the same notation \alpha seems to be used to refer two distinct variables (L101 and L148). Figure 1 includes notations such as CR_3 and H, but their definition is missing in the main document. While the paper provides a generative model for the successor map very briefly, the detailed method for the inference and the reasoning behind it is missing. Because of this insufficient description, it is very challenging to correctly understand the algorithm and assessing the novelty or significance of the proposed method. Is \alpha in L101 identical to the \alpha in L148 conditioning reward only on the arrival state s' seems unrealistic. w(s) in L98 and phi(s') in L101 are used without definition The main algorithm is in Appendix * After author response * I appreciate authors for addressing the clarity issues in the author response. The additional description in the author response was helpful in understanding the notations and algorithms. I hope the authors revise the final version to include the details that are missing in the current version. Now I understood the method and increased my rating. I agree with other reviewers that the proposed method is intuitive and not studied before, so worth being presented in NeurIPS.

[Author Response · NeurIPS 2019]

We'd like to thank the reviewers for their encouraging comments and helpful suggestions on how to best improve the
paper. We've tried to follows these as closely as possible and are excited to share the details below.
As suggested by both Reviewers 1&2 (**R1Q4&R2Q2**), we agree that it is much better to search through exploration
and learning rates when feasible. We now show results for SR learning rate $\alpha_{SR} \in [0.001, 0.005, 0.01, 0.05, 0.1]$ and
exploration $\epsilon \in [0., 0.05, 0.1, 0.15, 0.2, 0.25, 0.3, 0.35]$ for Experiments 1&2. For Exp.1 BSR performed best in all
40 settings. Best-performing results (for total steps taken in 4500 episodes) are presented in Table I. We also agree
with (**R1Q3&R2Q1**) that it is instructive to present per-episode statistics and trajectories, to make the results more
interpretable and tie them in with existing literature, and thank **R1&2** for these suggestions. Per-episode steps are
presented at the beginning, middle and end of training for respective best performing $\alpha_{SR}$ and $\epsilon$ (Fig. R1a), as well as
an 'average' setting of $\alpha_{SR} = 0.01$ and $\epsilon = 0.1$ (Fig. R1b) that shows how SSR and GPI don't have the capacity to
keep adapting in this case. For Exp.2 we also ran each parameter setting with or without CR offset based exploration
(**R2Q3**) for BSR, SSR and EW, and added $\epsilon = 0.4$. Total (Table II) and per-episode returns (Fig. R1c) of the best
parameter settings illustrate the effect of CR based exploration. Fig. 1Rd shows representative trajectories. We will

Table I

| Algo | Steps($10^3$) | $\epsilon, \alpha_{SR}$ |
|------|------|------|
| BSR-4 | $348 \pm 7$ | $0., 0.005$ |
| GPI-4 | $400 \pm 10$ | $0.05, 0.001$ |
| SSR | $398 \pm 9$ | $0.1, 0.001$ |
| KQ | $385 \pm 9$ | $0.05, 0.001$ |

Table II

| | CR offset exploration | | No CR offset exploration | |
|------|------|------|------|------|
| Algo | Returns($10^3$) | $\epsilon, \alpha_{SR}$ | Returns($10^3$) | $\epsilon, \alpha_{SR}$ |
| BSR-4 | $237 \pm 4$ | $0.25, 0.05$ | $105 \pm 4$ | $0.3, 0.1$ |
| SSR | $201 \pm 4$ | $0.25, 0.005$ | $87 \pm 4$ | $0.4, 0.001$ |
| EW-4 | $202 \pm 4$ | $025, 0.05$ | $78 \pm 5$ | $0.4, 0.005$ |
| Stored reward map | | | | |
| GPI-4 | $196 \pm 1$ | $0.35, 0.01$ | | |

Figure R1

integrate these results into Fig. 2 and follow up with Exp. 3. We apologize for omitting important related work form our
original submission (**R2Q1&Q4**), we now reference [1,2,3] when discussing limitations of SR for transfer in Sections 1,
4, and 6: In particular, we refer to the experiment in [1] showing the limited policy revaluation capabilities, and discuss
how [2] finds evidence of these limitations in human behaviour. We also outline important, qualitative differences
between the transfer experiments in [3] and our Experiment 1: in [3] the agent only has to learn four, partially disjoint
trajectories (connecting opposite corners of an open maze), resulting in much more limited ambiguity in optimal action
choice for most states. In signalled settings like our known quartile (KQ), or for agents with memory (e.g. RNNs) this
becomes a simple task. In our case most states can be part of a large number of optimal trajectories, with internal walls
providing for non-trivial dynamics. Studying performance across all these possible trajectories frames this problem
properly in terms of lifelong/multitask learning, and highlights which algorithms can adapt well across all tasks.
We apologise to **R3** for any lack of clarity in our presentation and model. We aimed to use notation standard in the
literature, but agree that it is crucial to clearly define all notation. We have reworked Section 3 to clarify the algorithmic
components, added detailed figure legends (e.g. for 1b: Dirichlet process mixture model of the convolved reward maps.
The model is defined by a base distribution H and concentration parameter $\alpha$, giving a random distribution over CR
maps.) and defined all notation. We now explicitly describe the Chinese restaurant process (CRP) view of the DP and
our particle filter to clarify the inference process. Briefly,the CRP gives a closed-form prior over the discrete latent
contexts at every step, each associated with both a CR map (e.g. $CR_3$ for context 3) and a successor map M. We base
our inference on observed CR values that depend on the current reward function and the policy the agent is following
according to the successor maps. This inference is intractable and we want to avoid specifying priors over M and
CR maps (priors normally defined by H). Amortizing the inference, we calculate a posterior over latent contexts by
combining the CRP prior with Gaussian likelihoods of the observed CR value given the value stored in the CR maps. We
update the sampled successor map M independently by TD update (L101), and then update the CR map at the end of the
episode. This allows the agent to both integrate evidence and refine the SR at each step, while updating the CR map of
the overall most likely context. Regarding **R3Q1**, the subscript was indeed omitted by mistake from the SR learning rate
$\alpha_{SR}$, we corrected this and included a definition in the text. Similarly (**R3Q3**), the state embedding $\phi(s)$ and the reward
vector **w** are now defined straight away, rather than later, and we point out that in the tabular case $\phi(s)$ is a one-hot
encoding of states and **w** the corresponding vector of reward per state. On the suggestion of **R1Q1** we also include a
detailed methodological description in the SI and the suggested reference. **R1Q2** raises an important question regarding
biological plausibility. In the tabular case, our algorithm relies only on TD updates, delta rule/Rescorla-Wagner type
update of the CR maps, and Bayesian filtering (filtering is implemented by neurons e.g in probabilistic population
codes or sampling based methods[4]). In the continuous setting we do rely on backpropagating the TD error through an
MLP, but see [5] for recent advances. We do believe this puts the algorithm firmly in the biologically plausible setting
compared to approaches relying on recurrent policy gradients or meta-learning with long-range backprop through time.
As suggested (**R1Q6**) there is also behavioural evidence that GPI is not a good fit in Section 5.2, as it both makes many
error trials, and takes longer to reach the goal, while animals (and BSR) have few error trials and tend to head straight
to a goal after initial mistakes. This is suggested by Fig.4h and discussed briefly, but we will include a more detailed
discussion. The correlation coefficient in section 5.1 (**R1Q5**) measures the correlation between the episode number and
the z-score difference ( Fig. 3c). It gives a measure of the transitioning from the old to the new maps. [1] Russek et al.
2017, [2]Momennejad et al. 2017, [3]Lehnert et al. 2017, [4]Kutschireiter et al 2017, [5] Whittington et al. [2019]

[Meta-Review · NeurIPS 2019]

The paper proposes an improvement of popular 'successor representation' approaches in reinforcement learning via a mechanism for maintaining and quickly updating a distribution over multiple successor maps. This innovation enables the model to adapt better to environmental changes such as different goals or reward structures. All three reviewers agree that this is a strong paper that should be accepted. I see no reason to contradict their opinion. While the reviewers were very positive, they did point out issues of clarity in the exposition, and we would like to remind the authors that their paper will reach a wider audience if they can make the presentation and explanation as clear and simple as possible in the camera ready version.